# Dynamic In Vitro Gastric Digestion Behaviour of Commercial Infant Formulae Made with Cow, Goat and Sheep Milk

**DOI:** 10.3390/foods13091286

**Published:** 2024-04-23

**Authors:** Xuan Song, Xin Wang, Mengxiao Yang, Alejandra Acevedo-Fani, Harjinder Singh, Aiqian Ye

**Affiliations:** Riddet Institute, Massey University, Private Bag 11 222, Palmerston North 4442, New Zealand; xuan.song@outlook.com (X.S.); m.yang5@massey.ac.nz (M.Y.); a.acevedo-fani@massey.ac.nz (A.A.-F.); h.singh@massey.ac.nz (H.S.)

**Keywords:** infant formulae, sheep milk, goat milk, gastric digestion, heat treatment, coagulation

## Abstract

There are a wide range of commercial infant formulae available on the market. These are made using milk from different species, such as goat, sheep, and cow. The different protein compositions of these milks and the process used during infant-formulae manufacture, such as heat treatment, may impact the digestion of nutrients. This study compared the effect of protein composition and heat treatment on the in vitro gastric digestion behaviour of commercial infant formulae made with cow, goat, and sheep milk using a dynamic infant human gastric simulator (IHGS). During the simulated dynamic gastric digestion, the goat milk infant formula (GIF) showed earlier signs of aggregate formation compared to cow milk infant formula (CIF) and sheep milk infant formula (SIF). In addition, the microstructures of GIF chyme showed fragmented and porous structures. On the contrary, CIF formed dense protein networks that trapped oil droplets, whereas SIF exhibited a microstructure of smooth oil droplets surrounded by fewer protein networks. The different aggregation behaviours and aggregate structures of the three infant-formulae chyme were related to their different protein compositions, especially the different casein compositions. Furthermore, the open fragile structure of GIF aggregates provided easier access to pepsin, allowing it to hydrolyse protein. The results from the present study provided some information to assist in understanding the coagulation and digestion behaviours of commercial infant formulae made from different species of milk.

## 1. Introduction

Dairy cattle and noncattle (such as sheep and goat) milk and milk products have been consumed over the past centuries [1,2]. Recently, there has been increasing interest in understanding the composition, physicochemical properties, and digestibility of cattle and noncattle milk [3,4,5,6]. Noncattle milk, especially goat and sheep milk, has increased in popularity among consumers due to its better digestibility and perceived health benefits [7,8,9].

Infant formula is a substitute for human milk, providing nutrients to neonates for their growth and development when breastfeeding is not available [10]. Compared to human milk, animal milk contains different quantities of protein, lactose, and minerals, such as higher casein-to-whey protein ratios [9,10,11,12]. Commercial infant formulae are designed to simulate the composition and biological properties of breast milk in order to provide high-quality nutrition for infants. In recent years, a wide range of commercial infant formulae made with milk from different species, such as goat and sheep milk, have come onto the market.

Milk from different species is known to have varied compositions, casein micelle characteristics, and physicochemical properties [9,11,13]. Compared to cow milk, goat milk contains lower proportions of α_s1_-casein and larger casein micelles, while sheep milk has higher total solid and mineral contents [5,9,14]. However, the composition of different milks is affected by many factors, such as feeding strategies, physiological factors, genetic factors, and environmental conditions [13,15]. Different protein and casein compositions lead to different enzymatic or acid coagulation behaviours and gels properties [6,16]. Goat milk is considered to be easier to digest due to the rough and soft coagulation formed under gastric conditions [14]. On the contrary, sheep milk forms a firm gel due to higher mineral and protein contents [17]. 

Milk protein digestion has been studied in both in vivo and in vitro models [18,19,20,21,22,23]. Most of the previous studies have focused on comparing the different compositions and physicochemical properties of cow, goat, and sheep milk, as well as the different components that affect gastric digestion behaviour. However, there are few studies on the effect of protein composition on the gastric digestion behaviour of infant formulae from different animal spices, especially the digestion behaviour of commercial infant formulae. Infant-formulae coagulation during digestion plays an important role in determining the gastric emptying rate, the rate of nutrient release, and the composition of material released into the small intestine [24,25,26]. Ye et al. (2019a) reported that the GIF formed smaller flocs of aggregated protein and oil droplets under gastric conditions, leading to faster protein digestion for goat milk infant formula (GIF) than for cow milk infant formula (CIF) [25]. The extent of the coagulation of protein and the size of flocculated oil droplets were dependent on the protein composition of the formulae. The casein-dominated CIF had a greater extent of protein aggregation initially during gastric digestion than the goat milk infant formulae, but had a lower rate of casein digestion.

Whey protein is commonly used to adjust the whey protein-to-casein level of infant formulae. Whey protein is classified as a fast dietary protein, remaining soluble in the stomach and rapidly passing into the small intestine [27]. The native structure of β-lactoglobulin (β-LG) is a globular structure that hides pepsin hydrolysis cleavage sites. On the other hand, α-lactalbumin (α-LA) can be hydrolysed by pepsin when the pH decreases below 4 [28].

Thermal treatment, including pasteurization, UHT, and extensive thermal treatment, is the most common unit operation in the manufacture of dairy products [29]. In infant-formulae manufacture, the liquid milk undergoes spray-drying, usually at 70–80 °C [29]. Heat treatment can induce β-LG unfolding and expose a free thiol group that initiates disulphide bond formation between adjacent whey proteins and casein micelles [30]. These conformational changes, induced by heating, can also lead to β-LG being more susceptible to hydrolysis by pepsin. Several studies have investigated the impact of heat treatment on the different structures of clots formed during gastric digestion [19,31,32,33,34]. Ye et al. (2019b) observed different clot structures for heated and unheated skimmed milk during in vitro gastric digestion. The clots formed by heated milk showed more open knitted networks with larger pores [35]. In addition, Pan et al. (2021) reported that the heated and homogenized sheep milk formed a more fragile curd structure than the untreated milk [36]. However, these studies were carried out on fresh milk, and there is insufficient information available on infant formulae.

This study aimed to compare the coagulation and digestion behaviours of commercial infant formulae made from cow, goat, and sheep milk under in vitro infant gastric conditions. Moreover, the aggregation behaviour of heat-treated (90 °C for 5 min) cow-, goat-, and sheep-based infant formulae was investigated in comparison with their unheated infant formulae under gastric conditions. The experimental work was carried out using a dynamic in vitro infant human gastric simulator (IHGS).

## 2. Materials and Methods

### 2.1. Materials

Commercial infant formulae for infants aged from 6 to 12 months and made from sheep milk, goat milk, and cow milk were purchased from a local supermarket in New Zealand. Their chemical compositions are shown in Table 1. Pepsin from porcine gastric mucosa (EC 3.4.23.1; 380 units/mg solid) was purchased from Sigma-Aldrich Co. (St. Louis, MO, USA). Milli-Q-treated water was used for all the experiments. All other reagents used were of analytical grade and were purchased from Sigma-Aldrich Co. (St. Louis, MO, USA) or BDH Chemicals (BDH Ltd., Poole, UK), unless otherwise specified.

### 2.2. Preparation of Liquid Infant-Formulae Samples

Then, 100 g (1.6% (*w*/*w*) protein) reconstituted infant-formula samples were prepared by dissolving each infant formula in Milli-Q water, with continuous stirring for 30 min at 50 °C. The reconstituted samples were then warmed at 37 °C for in vitro digestion. In a separate experiment, the reconstituted infant formulae were heated at 90 °C in a water bath, held for 5 min, then immersed in ice water, and warmed at 37 °C prior to gastric digestion. The formula was composed of 1.6 g of protein per 100 g of formula, as measured by the Kjeldahl method. They contained 3.2, 3.3, and 2.4 g of fat for 100 g of sheep milk-based, goat milk-based and cow milk-based formula, respectively, as measured by the Mojonnier method.

### 2.3. In Vitro Dynamic Gastric Digestion

In vitro infant gastric digestion was studied in a dynamic Infant Human Gastric Simulator (IHGS) [25]. The simulated gastric fluid (SGF) consisted of two separate solutions: SGF-HCl (salts and HCl; pH 2) and a pepsin solution (pepsin and CaCl_2_). SGF-HCl (salts and HCl; pH 2) was prepared as described by Minekus et al., (2014) [37] with slight modifications. The activity of pepsin in the mixture of SGF and reconstituted infant formula was 80 units/mL/kg body weight (i.e., 7.5 kg for 6-month-old infants) [38].

The gastric digestion of the infant formula followed a procedure reported previously, with some slight modifications [25]. The IHGS was prewarmed and maintained at 37 ± 0.5 °C by two internal heaters and a thermostat. In each experiment, 100 mL of reconstituted infant formula was warmed at 37 °C prior to digestion. To simulate the fasting conditions in infant stomachs, the prewarmed sample was mixed with 9 mL of basal SGF [i.e., 7.2 mL of SGF-HCl (pH 2) and 1.8 mL of pepsin solution] and was then poured into the IHGS [39]. To simulate gastric secretion, the SGF-HCl and the pepsin were pumped gradually into the gastric chamber separately at flow rates of 0.4 and 0.1 mL/min, respectively. To mimic infant gastric emptying, 22 mL of gastric digesta was sampled at 20 min intervals via a silicone tube attached to the bottom of the stomach chamber during 180 min of gastric digestion [40]. The collected digesta sample was filtered through a 1 mm pore mesh, which prevented the emptying of larger particles, in order to simulate the effects of gastric sieving [41]. The gastric contraction frequency of the infant HGS was 3 contractions/min, which aimed to mimic the natural contraction of the infant stomach [42].

### 2.4. pH Measurement

The pH of the reconstituted infant formula samples and their digesta during gastric digestion were measured using a pH meter (ATC, Stand-EC-PH700-42s, EUTECH INSTRUMENTS, Singapore). The initial pH (T0) was defined as the pH of the mixture of reconstituted infant formula and basal SGF. During the 180 min digestion, the pH of emptied digesta at each time point were represented by the pH value in the IHGS, as the roller movement of the IHGS prevented easy access to the latex stomach chamber.

### 2.5. Particle Size Measurement

We measured the particle size distribution of the undigested infant formula and emptied digesta using a Mastersizer 2000 (Malvern Instruments, Malvern, Worcestershire, UK). Refractive indices of 1.46 (with an absorbance value of 0.001) and 1.33 were used for the dispersed phase and aqueous phase, respectively. The particle size values were characterized by the volume-weighted average diameter (*d*_4,3_). To better understand the effect of flocculation, in a separate experiment, volumes of initial and gastric chyme samples were dispersed in 2% (*w*/*v*) SDS and 50 mM of EDTA solution (1:4, *v*/*v*) to dissolve protein aggregates and casein micelles. The mixture was gently mixed for at least one hour before we performed particle size measurement. Mean particle diameters were calculated as the average of triplicate measurements performed on individual samples. All measurements were carried out in three replicates.

### 2.6. Confocal Laser Scanning Microscopy

The microstructures of the samples formed from different formulae and the gastric chyme of infant formulae during in vitro gastric digestion were observed using confocal laser scanning microscopy (Leica ZEISS LSM 900 with Airyscan 2, Leica microsystems, Heidelberg, Germany). A 400 µL sample was mixed with 20 µL of 0.1% (in acetone, *w*/*v*) Nile Red and 10 µL of 1.0% (*w*/*v*) Fast Green and were examined within 30 min. For the gastric chyme, the samples were collected at 80, 100, and 180 min during in vitro gastric digestion and were stained and observed immediately. Similarly, 400 µL of liquid chyme was mixed with 20 µL of 0.1% (in acetone, *w*/*v*) Nile Red and 10 µL of 1.0% (*w*/*v*) Fast Green. Nile Red (0.1% in acetone, *w*/*v*), excited with an argon laser at a wavelength of 488 nm, was used to stain the oil phase. Fast Green (1.0%, *w*/*v*), excited with a helium–neon laser at 633 nm, was used to stain protein. The stained samples were then placed onto double-concave microscope slides. Each sample was prepared in duplicate, and multiple fields were viewed, with only representative micrographs being presented. Images were stored with 1024 × 1024 pixel resolution using digital image processing software (ZEISS Zen 3.1).

### 2.7. Sodium Dodecyl Sulfate Polyacrylamide Gel Electrophoresis (SDS-PAGE)

The time-dependent hydrolysed protein present in infant formula and their emptied digesta were assessed via SDS-PAGE (15% acrylamide gel). Samples collected from 20, 40, 80, 100, 120, and 160 min were observed. Different amounts of sample were mixed with 800 μL of sample buffer (25% glycerol, 0.5M Tris-HCl, pH 6.8, 10% SDS, 0.1% Bromophenol Blue, 5% β-mercaptoethanol) to achieve equal protein content (0.1%). All solutions were heated in a 90 °C water bath for 10 min. They were then cooled to room temperature and 10 μL was loaded onto a resolving gel. Electrophoresis analysis was carried out at a constant voltage of 125 V for approximately 90 min until the bromophenol blue dye line reached the bottom of the gel. We stained the gel for 40 min with a Coomassie Brilliant Blue R-250 solution, and then destained it with a destaining solution overnight.

### 2.8. Chemical Composition Analysis

The total protein (total nitrogen × 6.38) contents of the infant formulae and their emptied digesta were determined using the Kjeldahl method, and fat contents were determined using the Mojonnier ether extraction method [43]. The emptied digesta, collected at 20, 40, 80, 100, 120, and 160 min, were assessed to determine protein content, and the fat content of digesta obtained from 60, 140, and 180 min was determined.

### 2.9. Statistical Analysis

We repeated all experiments a minimum of two times using freshly prepared samples. A repeated-measures two-factor ANOVA model with in vitro replication as the experimental unit was performed for the PH, particle size, protein, and fat content of infant formulae and their empty digesta using the MIXED model procedure of the SAS statistical software (SAS/STAT version 9.4; SAS Institute Inc., Cary, NC, USA). The statistical linear mixed model included the sample number (1, 2, or 3), duration of testing (0 to 180 min), and interaction as fixed effects, whereas replication was a random effect. The most appropriate covariance structure for the mixed models was selected after fitting the models using the restricted-maximum-likelihood method and comparing them using the log-likelihood ratio test. 

## 3. Results

### 3.1. Protein Composition of Infant-Formula Samples

The protein composition of sheep milk infant formula (SIF), GIF, and CIF was analysed via SDS-PAGE under reducing conditions. Figure 1 shows that the three infant formulae contained different protein compositions, especially in terms of the proportions of caseins. β-Casein was the most visible band in SIF, and both α_S1_-casein and α_S2_-casein brands were not observed. GIF exhibited a clear α_S2_-casein band. However, there was no visible α_S1_-casein band, and β-casein was the predominant casein in GIF. On the contrary, the α_S1_-casein band was clearer in CIF, with a similar proportion to β-casein, while α_S2_-casein was very faint in CIF. The κ-casein band showed similar intensity in all samples. The observed SDS-PAGE pattern of caseins in the three IFs agrees with the previous reports on these species milk in that there is a lower level α_S1_-casein in goat milk than in other ruminants’ milk, while cow milk contains a higher level of α_S1_-casein [5,8]. For the casein fraction of sheep milk, previous reports present contradictory findings; α_S1_-casein has been reported to range from 5.3% to 39.9% of the total casein [44]. Additionally, casein concentrations vary between different sheep breeds [9]. The whey-to-casein ratio of SIF, GIF, and CIF agreed with the label claim of containing 60% whey protein, which is typical of a whey protein-dominant infant formula.

### 3.2. Simulated Gastric Digestion Behaviour

#### 3.2.1. pH Changes during Gastric Digestion

The changes in the pH values of empty digesta from SIF, GIF and CIF were monitored for 180 min of dynamic in vitro infant gastric digestion (Figure 2). There was no significant difference between the three samples (*p* > 0.05). The pH profile of three samples showed a decreasing trend with the constant addition of SGF for 180 min. The initial pH values of SIF, GIF, and CIF with fasting SGF were 6.42 ± 0.07, 6.54 ± 0.02 and 6.74 ± 0.02, respectively; during the first 60 min, the pH showed a slight decrease, but from 60 min to 180 min, the pH decreased to around 3.

Comparing the heat-treated (90 °C, 5min) samples with unheated ones (Figure 2), the pH values of heated SIF were higher than those of unheated SIF at 120–140 min digestion (*p* < 0.05). In the period of 120–180 min, the pH of heated SIF dropped from 4.85 ± 0.11 to 2.87 ± 0.17, while the pH of unheated SIF decreased from 4.20 ± 0.19 to 2.59 ± 0.23. The overall pH profile of heated SIF showed a slower decrease than that of the unheated SIF. Similarly, the pH of the heated sample of GIF was 5.80 ± 0.15 at 100 min and this gradually decreased to 2.90 ± 0.13 at the end of digestion, while the pH of the unheated sample declined from 5.68 ± 0.05 to 2.67 ± 0.16 in the same period. However, the pH of heat-treated CIF showed no significant difference from the unheated CIF (*p* > 0.05). This result is in agreement with a study comparing sheep whole milk with heated, homogenized sheep milk [36]. The slow decrease in pH of heated milk could be related to the denaturation of whey protein associated with casein micelles, forming more open, structured aggregates under gastric conditions [34].

#### 3.2.2. Microstructure of Chyme in the Stomach

We observed the microstructure of three infant formulae and their chyme in the stomach at 80 min and 100 min of digestion via confocal laser scanning microscopy (CLSM) (Figure 3). At 0 min, three infant formulae showed uniformly dispersed oil droplets of similar size, without any coagulation or flocculation. At 80 min of digestion, where pH was around 6.2–6.3, protein coagulation was observed. For GIF and CIF, the oil droplets were embedded in the protein coagulum, whereas in SIF, the protein coagulum was less intense, and more individual oil droplets could be observed.

There were no changes in the microstructures of samples at 100 min digestion when pH was around 5.5, compared to that at 80 min. The largest flocs and particle sizes of oil droplets in three samples were observed at 180 min (pH~2.6). This indicated that the coalescence of oil droplets occurred at a later stage of gastric digestion for all infant formulae. In general, there was more droplet coalescence than aggregation, and the individual oil droplets were more uniform in size in SIF. In contrast, GIF showed dense, coarse, and porous aggregates. The oil droplets were of various sizes and incorporated into the coagula. For CIF, more protein aggregates were observed, along with various-sized oil droplets trapped in the aggregates. Overall, SIF showed smooth oil droplets with fewer protein networks. CIF exhibited dense protein networks that trap oil droplets. By contrast, GIF formed irregular aggregates with porous and coarse textures.

As shown in Figure 3, no significant difference was observed between heated and unheated treated SIF. Both sets of samples showed uniformly dispersed droplets at 0 min. Aggregations were then observed at 80 min. The oil droplets merged into larger round-shaped droplets, which were embedded in protein flocculation. From 100 min, the size of oil droplets and protein aggregates increased with digestion time, and they were the largest at the end of digestion. On the contrary, heat-treated CIF and GIF appeared in different microstructures compared to the unheated ones. A small number of aggregates with dense structures were observed in heated CIF and GIF at 80, 100, and 180 min, and the aggregates were smaller than the unheated samples. The more open structures in heated CIF and GIF were in a line with a previous study of whole bovine milk [45]. Ye et al. (2017) found that the clot structures of the heated milk sample contained more open pores [45]. Moreover, no oil droplet coalescence was observed in heated CIF or GIF. 

#### 3.2.3. Changes in Particle Size during Gastric Digestion

Figure 4A shows the particle sizes of the unheated IFs chyme in the stomach and that of the chyme that was dispersed in SDS and in the EDTA solution. The *d*_4,3_ values of all IFs before ingestion by the HGS were similar (~0.5 ± 0.1 μm). The average sizes of all samples gradually increased during digestion. The *d*_4,3_ values for CIF and GIF were around 40 μm, and the *d*_4,3_ value of SIF was around 17 μm after 180 min of digestion. Particle size was reduced markedly in SDS and in the EDTA solution, indicating the droplet flocculation of IFs during digestion. GIF and CIF showed more droplet flocculation than SIF. In addition, the size of IFs in SDS and in the EDTA solution increased with digestion time (Figure 4A), indicating that some coalescence of oil droplets in the IFs occurred during digestion.

The changes in the average particle size of the emptied digesta showed different trends. As shown in Figure 4B–D, the average diameters of the three emptied digesta did not change until 60 min (*p* > 0.05), and increased slightly at 100 min. After 140 min of digestion, the average diameters of emptied GIF digesta increased to be higher than the other two samples and increased dramatically at the end of the digestion. The *d*_4,3_ value of GIF increased from 8.59 ± 1.77 μm at 140 min to 41.75 ± 5.25 μm at 180 min, whereas the *d*_4,3_ value of emptied digesta of CIF raised from 5.28 ± 1.37 μm to 19.71 ± 6.08 μm, and the *d*_4,3_ value of SIF emptied digesta increased from 5.85 ± 0.75 μm to 16.43 ± 3.54 μm during the same period. Overall, similar trends were found in the emptied digesta of three samples, but the average particle size followed the order: GIF > CIF > SIF (*p* < 0.05).

There were subtle differences in particle size between the heated and unheated samples of emptied digesta during digestion. The heated emptied digesta of SIF followed the same trend as unheated SIF (*p* > 0.05). As shown in Figure 4B, the average diameters of both heated and unheated emptied SIF digesta stabilized at ~0.5 μm till 60 min, after which both increased significantly with digestion time. From 100 to 180 min of digestion, unheated SIF increased from 1.60 ± 0.13 μm to 16.43 ± 3.54 μm, while heated SIF rose from 1.29 ± 0.17 μm to 17.12 ± 3.54 μm. On the contrary, the *d*_4,3_ values of both heated and unheated emptied digesta of GIF remained at the same levels before 100 min of digestion (*p* > 0.05). Thereafter, the unheated emptied GIF digesta had a dramatic increase from 8.59 ± 1.77 μm at 140 min to 41.75 ± 5.25 μm at 180 min, whereas the heated GIF increased from 4.93 ± 6.08 μm at 140 min to 6.01 ± 0.91 μm at 180 min. Similarly, the average diameter (*d*_4,3_) of heated emptied CIF digesta followed the same trend as unheated CIF, except the unheated CIF increased sharply at 180 min (*p* < 0.05) (Figure 4D). The average size at 180 min was 19.71 ± 6.08 μm versus 11.46 ± 3.01 μm for unheated and heated CIF emptied digesta, respectively.

#### 3.2.4. Empty Rate of Protein and Fat during Gastric Digestion

The protein contents of emptied digesta, derived from different IFs as a function of digestion time, are presented in Figure 5(A-P,B-P,C-P). Because of the gradual addition of SGF, samples were diluted continuously with the progression of the digestion process. Assuming that the sample was distributed homogeneously in the stomach throughout the entire digestion process, the diluted concentration of protein and fat over digestion time was calculated and is shown as the “dilution curve” in Figure 5.

All samples went through a similar trend (*p* > 0.05). In the first 20 min of digestion, the protein content of all samples decreased slightly, following the trend of the dilution line. From 40 to 100 min, the protein content gradually declined but remained higher than the expected based on dilution. After 100 min, the protein content kept decreasing to a level that was slightly lower than the dilution line. Having protein contents above the dilution line in the 40–100 min period indicated that there was protein aggregation and precipitation in the samples, which led to more protein being released from stomach. After 100 min, where pH reached <5, the protein contents in the digesta were lower than the dilution value. This indicates that the protein hydrolysis conducted by pepsin occurred in the stomach.

As shown in Figure 5(C-P), no significant difference (*p* > 0.05) was observed between heated and unheated CIF at each time point of digestion. However, GIF and SIF protein contents exhibited differences at certain time points (Figure 5(A-P,B-P)). Compared to the protein content of heated GIF, which almost followed the dilution line except for 40 min, the protein content of unheated GIF was higher than the dilution line at 40 and 80 min, but then decreased to a level slightly under the line at 120 and 160 min. There were significant differences between heated and unheated GIF protein contents at 80 and 160 min. At 80 min, the unheated GIF contained higher protein content than that of the heated GIF, but it decreased to a level lower than that of the heated GIF at 160 min. Similar trends were found in heated and unheated SIF. The protein contents of both samples were higher than the dilution line and dropped to a level below the line after 100 min. However, the protein content of heated SIF was lower than that of unheated SIF before 100 min, but then it shifted to a higher value compared to the unheated SIF until 160 min. Significant differences were observed at 80 and 120 min (*p* < 0.05).

The fat contents of emptied digesta, derived from heated and unheated CIF, GIF and SIF as a function of digestion time, are presented in Figure 5(A-F,B-F,C-F). At the early stage of digestion, the fat content mostly decreased following the dilution trend, but it sharply dropped to a level under the dilution line from 60 min to 140 min. This could be attributed to the extensive flocculation of oil droplets, resulting in creaming on the top of stomach, which trapped more fat in the stomach.

As shown in Figure 5(A-F,B-F,C-F), the fat content of heated and unheated CIF was above the dilution line, but sharply dropped under the dilution line around 80 min. At the end of the digestion, the fat content in the emptied digesta of the heated sample was higher than that in the unheated one. Regarding the fat content of GIF, the fat content of heated samples was higher than that of the unheated samples after 60 min of digestion; this was significantly higher than that level seen in the unheated emptied digesta at 140 min (*p* < 0.05). Similar trends were observed in the fat content of the heated and unheated SIF emptied digesta, expect at 180 min, where the value of the heated sample was higher than that of the unheated sample (*p* < 0.05). Overall, the fat content of heated and unheated emptied digesta of three infant formulae, regardless of whether they were heated or unheated, showed similar trends. Also, the fat contents of three heated emptied digesta from three heated samples were all higher than those of the corresponding unheated samples at 180 min.

#### 3.2.5. Hydrolysis of Protein during Gastric Digestion

The protein hydrolysis of emptied digesta of SIF, GIF, and CIF during infant gastric digestion was analysed via SDS-PAGE under reducing conditions (Figure 6). All IFs showed that the κ-casein band disappeared after 20 min of digestion, when the pH was 6.76 ± 0.02, whereas the other protein bands (especially α_s1_-casein, α_s2_-casein, β-casein, β-LG and α-LA) remained unchanged for the intensities at 0, 20, 40 and 80 min. The result agrees with the previous studies, which showed that κ-casein is hydrolysed by pepsin in preference to other caseins [25,33]. Also, a 15-kDa band was observed at 20–100 min, indicating that κ-casein was hydrolysed and formed para-κ-casein. Beyond 100 min, the casein bands decreased, and they disappeared from 120 min onwards from all IF samples. At 160 min, even the β-LG band disappeared from SIF, but it remained in GIF and CIF, although the band’s intensity was significantly lower (Figure 6). The disappearance of casein bands was attributed to the easy hydrolysis of casein by pepsin. It is well known that whey protein, especially β-LG in the native structure, is less sensitive to pepsin [45]. β-LG resists hydrolysis by pepsin and remains soluble in the stomach. Only β-LG denatured via pre-treatment, such as heating treatment, leading to aggregation and hydrolysis by pepsin [31]; this may explain the different protein hydrolysis patterns among SIF, GIF, and CIF in the present study. The commercial SIF might have been treated via different processes (higher heat treatment intensity), which may have led to a higher degree of denaturation of the whey protein.

When comparing GIF and CIF, the casein bands of GIF almost disappeared at 100 min, but the casein bands of CIF were still observed. This indicated that the protein in GIF may be more susceptible to hydrolysis than that of CIF. This is in agreement with the previous studies showing that GIF forms porous and soft coagulum, which is easier to digest in the stomach [25,46,47]. The whey protein band intensities exhibited a similar trend in terms of GIF and CIF. The result aligns with the previous observations on pasteurized milk clots in cow and goat milk [6].

Compared to unheated samples (Figure 6, on the left-hand side), heated (Figure 6, on the right-hand side) GIF and CIF exhibited different patterns of protein hydrolysis. In heated GIF and CIF, casein bands reduced in intensity gradually until 120 min, when no more casein bands could be observed. This indicated that casein hydrolysis started much earlier in heated CIF and GIF. The results are in agreement with the report of Miranda and Pelissier (1987), which showed that heat treatment improved the casein hydrolysis and gastric emptying rate [32].

The differences in whey protein hydrolysis between the heated and unheated emptied digesta of CIF and GIF were even more obvious. Both α-LA and β-LG bands in the heated sample showed decreased intensities with increasing digestion time. At 160 min, no whey protein bands could be observed in the heated sample, while β-LG brands were still present in unheated GIF and CIF. Ye et al. (2017) also observed that both casein and whey proteins hydrolysed slower in unheated milk compared to heated milk [33].

On the contrary, both unheated and heat-treated SIF showed similar trends (Figure 6 SIF). Casein bands in both heated and unheated SIF became faint from 100 min, while whey protein bands decreased in intensity from 120 min onwards. Casein bands could not be seen at 120 min, and whey protein could not be observed at 160 min in both samples. At the end stage of digestion, there should be many peptides evacuated from the stomach, which are too low in molecular weight to be detected using SDS-PAGE.

## 4. Discussion

The present study showed that there was no firm clot formed in sheep, goat, or cow commercial infant formulae during gastric digestion in the infant human gastric simulator. This contrasts what has been observed during the digestion of these milks [36,45,48], in which casein micelles coagulate and form a firm clot, a process induced by the presence of pepsin and low pH levels under gastric conditions. This could be attributed to different ratios of casein and whey proteins and the processing steps involved in the manufacture of infant formulae. However, different structures of aggregates/flocs in the chyme within the stomach could be observed. SIF formed smooth oil droplets with fewer protein networks. GIF formed irregular aggregates with porous and coarse textures, whereas CIF exhibited dense protein networks. The difference in structures among three infant-formulae chyme could be attributed to the differences in their protein composition, especially the different constituents of caseins. As shown via SDS-PAGE, the protein compositions of three infant formulae were different; GIF and SIF contained higher proportions of β-casein and lower proportions of α_s_-caseins, while CIF contained similar amounts of β-casein and α_s_-caseins.

According to Roy et al., (2020), the formation of partial and fragile aggregates could be related to the β-casein-to-α_s_-casein ratio in milk [49]. Li-Chan and Nakai (1988) modified casein using rennin to increase the β-casein fraction relative to α_s1_-casein in bovine milk in order to match the casein composition of human milk [50]. In an in vitro study of the acid coagulation of caseins, the results showed that the rennin-modified cattle milk, which contains a higher β-casein-to-α_s_-caseins ratio, had a similar coagulation behaviour to human milk. Under the conditions of pH 2 and pH 4, rennin-modified casein formed looser and less structured clumps, whereas the coagulum of bovine casein was large and dense. In addition, the previous studies of the in vitro digestibility of skimmed cow, goat, and sheep milk reported that non-cattle milk proteins, especially goat milk proteins, have a higher susceptibility to hydrolysis by pepsin due to faster hydrolysis [25,46,51,52]. This aligns well with the findings of this present study on the protein hydrolysis of three infant formulae. The SDS-PAGE patterns indicated that casein hydrolysis was faster in GIF than in SIF or CIF.

The structure of protein aggregates could influence the rate of protein hydrolysis and thus the digestion behaviour, but few differences were detected in the emptied digesta of three infant formulae. The result agrees with the study of Maathuis et al., (2017) who reported that GIF and human milk showed faster protein digestion compared to CIF in the early stage of digestion, but slower rates at the end stage, which may be due to their different protein compositions leading to different clotting behaviours in the stomach; this could result in different gastric emptying rates [47]. However, the results obtained by analysing the bioaccessible nitrogen and amino acids suggested that the protein quality and digestibility of GIF, CIF, and human milk were similar. Another reason for this could be that the infant formulae tested in the present study are whey-dominated products and the reconstituted milk has a low protein concentration, following feeding guidance (1.6% *w*/*w*). The low protein concentration of whey-protein-dominated infant formula leads to a low casein concentration under gastric conditions. In such cases, the different aggregate structures may only be obtained in the chyme during digestion, and the differences in emptied digesta can only be observed at the late stage of digestion when the pH decreases to below the isoelectric point of casein (pI 4.6) [30].

The present results indicate that heat treatment influenced the formation of protein aggregates and oil droplet flocculation during gastric digestion (Figure 4). The particle size distributions and microstructures of heated CIF and GIF were different to those of the unheated samples (Figure 4). The microstructures of chyme taken from heated samples showed more crumbled and porous structures. Also, the heated IFs showed earlier initial aggregation than the unheated IFs. In addition, the fat contents in the emptied digesta of heated IFs were lower than those of unheated Ifs at early stages of digestion, whereas they were higher than those of unheated IFs at the late stage of digestion. This suggests that the heated IFs may have a slower release of fat from stomach compared to unheated IF samples, which might be attributed to greater creaming of fat globules from the more fragile aggregate of heated samples. The different digestion behaviours could be ascribed to the denaturation of whey proteins induced by heat treatment, and the denatured whey protein could associate with casein micelles [33,45,53,54]. The infant formulae tested in the present experiment were whey protein-dominant emulsions. In the native structure, β-LG is resistant to enzymatic hydrolysis as its compact globular structure hides its cleavage sites [27]. According to Singh and Have (2003) and Ye et al., (2019), whey protein mostly denatures when heated up to 90 °C [35,54]. The cleavage sites of unfolded whey protein were explored and found to be associated with disulphide bond aggregations between adjacent whey proteins and between casein micelles and whey proteins [30,53,55]. Wang et al. (2018) reported that no aggregation was observed in whey protein isolate (WPI) during gastric digestion, whereas heated WPI aggregated in the early stage of digestion [21]. In addition, according to the previous studies, the denaturation of whey proteins led to whey proteins being involved in the formation of milk clots and accelerated gastric emptying [33,56]. This is consistent with the observations of comparative SDS-PAGE patterns of heated and unheated infant formulae in the present stud, where both casein and whey protein in heated CIF and GIF showed higher digestion rates in IHGS. There was no significant difference between heated and unheated SIF, which could be because SIF had already undergone more intense heat treatment during the manufacturing process. 

## 5. Conclusions

Infant formulae made from the milk of different species vary in their dynamic in vitro gastric digestion behaviours, which could be related to the different physicochemical properties and various protein compositions of milk of different species, such as the different casein compositions, casein micelle sizes, and structures in cow, goat, and sheep milk. However, gastric digestion is not only influenced by its different protein composition from different animal milk, but also impacted by the various processes such as heating treatment. The various in-processing treatments applied to commercial IF products may erase the effect of the native differences in the composition of species milk on digestion. Overall, the results from this experiment offer valuable insights into understanding the coagulation and digestion behaviours of infant formulae made from cow, sheep, and goat milk.

## Figures and Tables

**Figure 1 foods-13-01286-f001:**
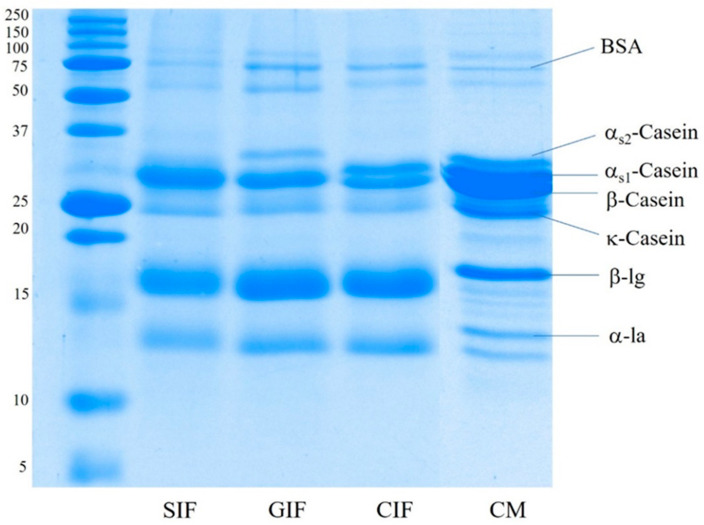
SDS-PAGE pattern under reducing conditions of the sheep infant formula (SIF), goat infant formula (GIF), and cow infant formula (CIF). CM: cow milk.

**Figure 2 foods-13-01286-f002:**
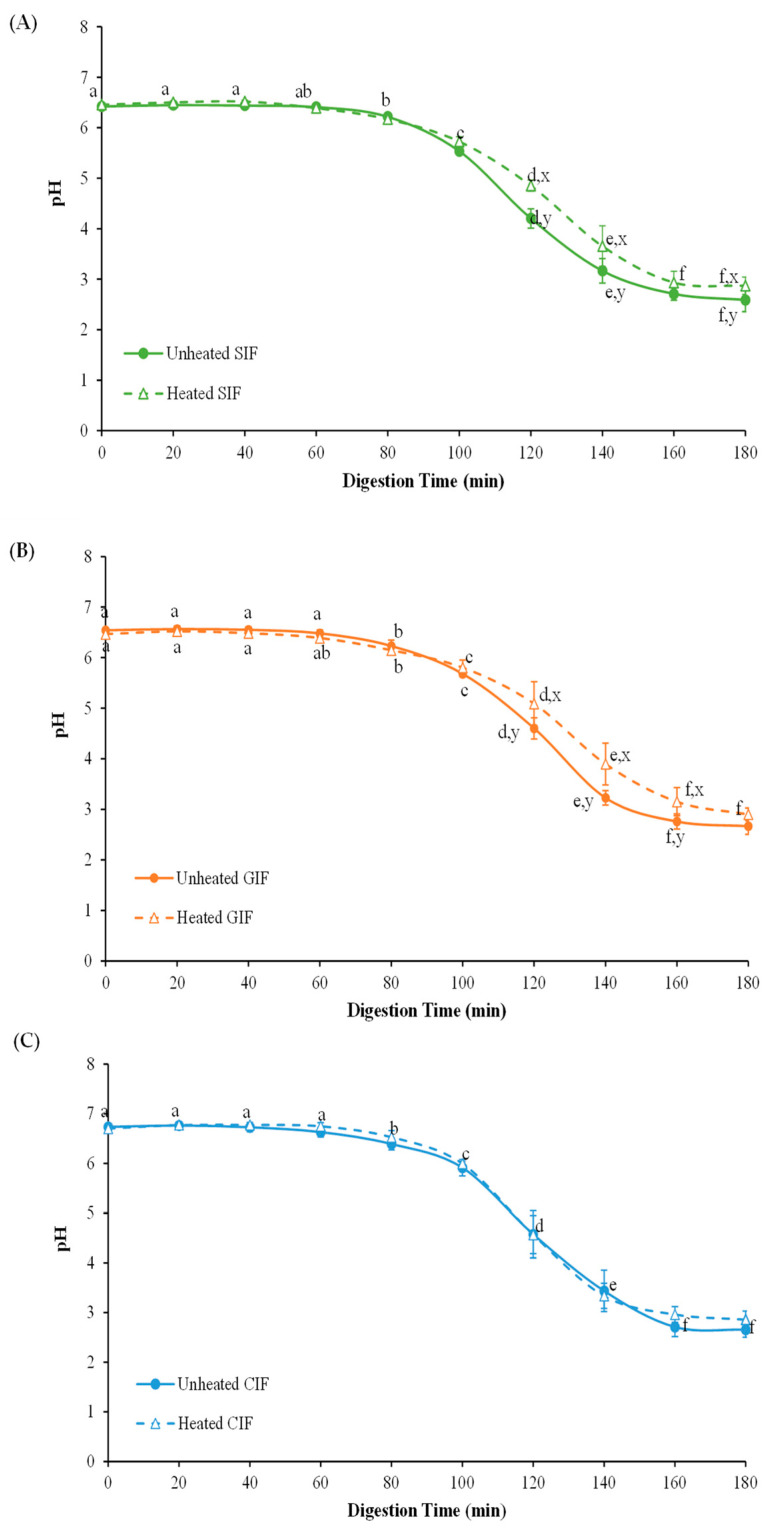
PH profiles of unheated and heated sheep (**A**), goat (**B**), and cow (**C**) milk infant formulae with 1.6% (*w*/*w*) protein during gastric digestion in IHGS. Values with no letter in common (a–f) represent significant differences (*p* < 0.05) within a certain treatment (with or without heating) across different digestion times. Values with no letter in common (x, y) represent significant differences at a certain digestion time across different treatments (with or without heating). If no letter is listed, there are no significant differences.

**Figure 3 foods-13-01286-f003:**
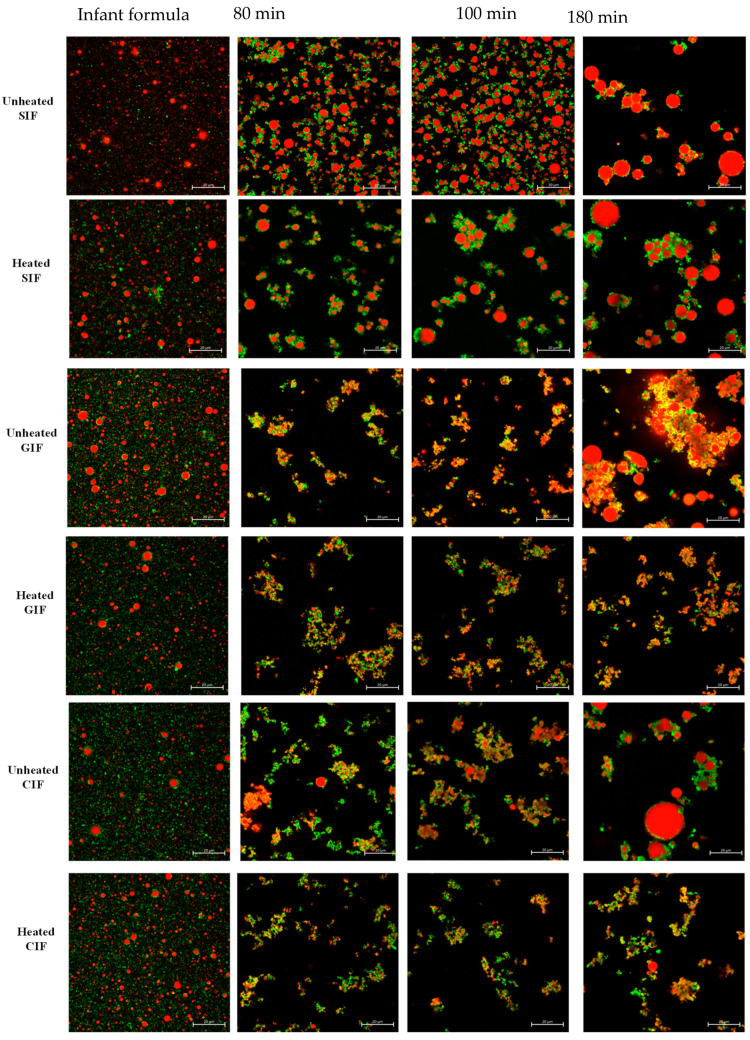
Confocal laser scanning microscopy (CLSM) images of unheated and heated SIF, GIF, and CIF chyme during gastric digestion in IHGS. Red shows the fat, and green shows the protein. The scale bar in all images is 20 µm.

**Figure 4 foods-13-01286-f004:**
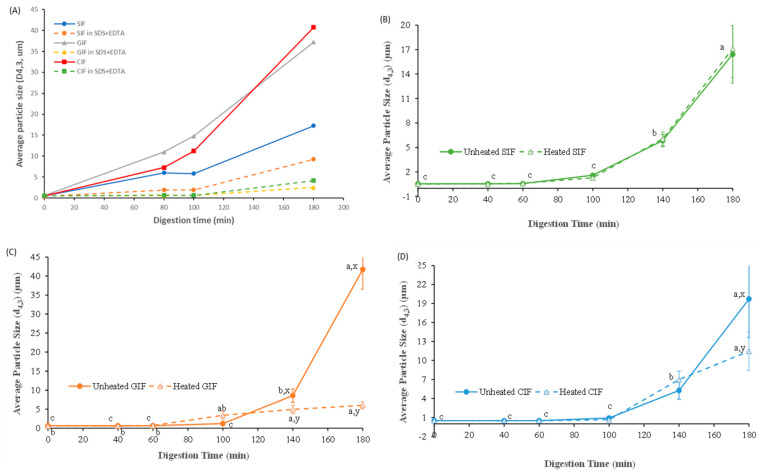
Average particle size (*d*_4,3_, μm) of the unheated IFs chyme in stomach and that of the chyme that was dissolved in SDS and in the EDTA solution (**A**). Changes in volume_weighted average diameter (*d*_4,3_, μm) of unheated and heated emptied SIF (**B**), GIF (**C**), and CIF (**D**) digesta during gastric digestion in IHGS. Values with no letter in common (a, b, c) represent significant differences (*p* < 0.05) within a certain treatment (with or without heating) across different digestion times. Values with no letter in common (x, y) represent significant differences at a certain digestion time across different treatments (with or without heating). If no letter is listed, there were no significant differences.

**Figure 5 foods-13-01286-f005:**
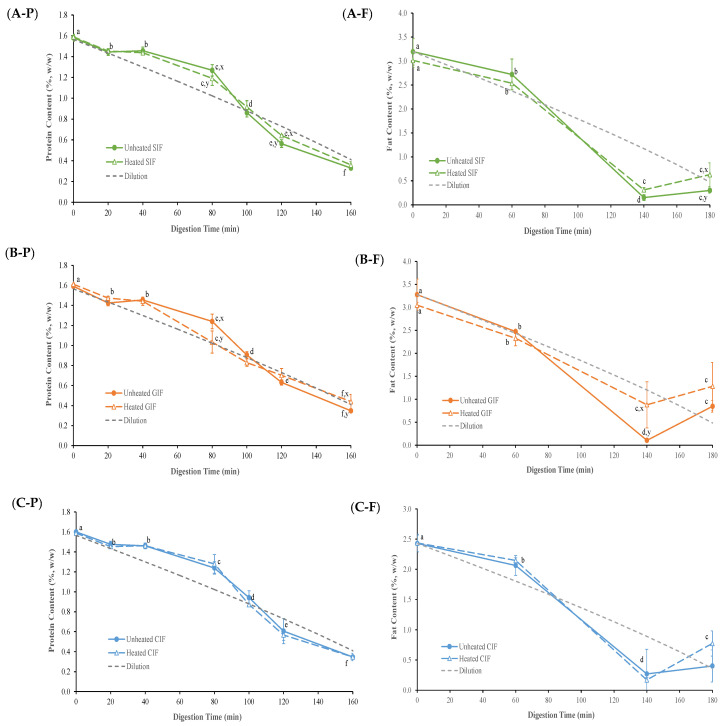
Protein contents in the emptied digesta of unheated and heated SIF (**A-P**), GIF (**B-P**), and CIF (**C-P**) during gastric digestion in the IHGS. Fat contents in the emptied digesta of unheated and heated SIF (**A-F**), GIF (**B-F**) and CIF (**C-F**) during gastric digestion in the IHGS. Values with no letter in common (a–f) represent significant differences (*p* < 0.05) within a certain treatment (with or without heating) across different digestion times. Values with no letter in common (x, y) represent significant differences at a certain digestion time across different treatments (with or without heating). If no letter is listed, there were no significant differences.

**Figure 6 foods-13-01286-f006:**
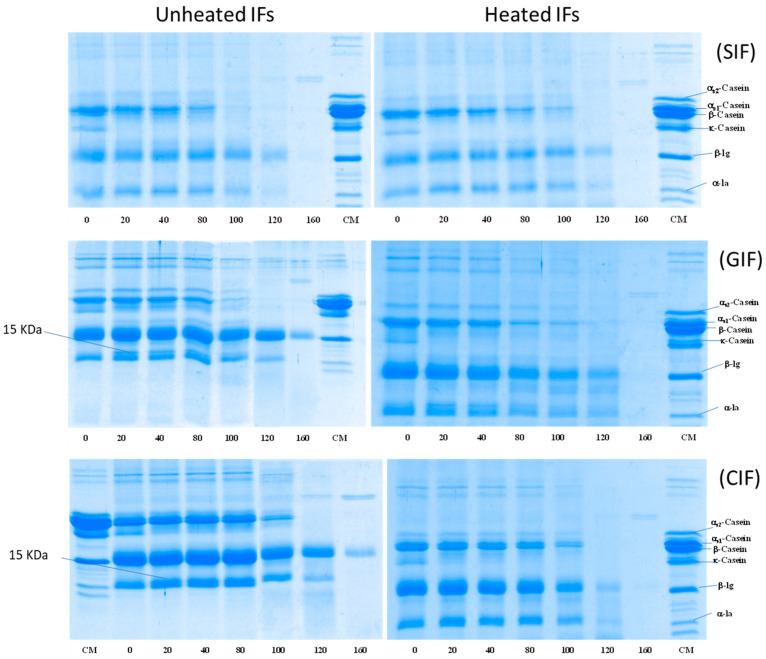
SDS-PAGE pattern under reducing conditions of emptied digesta obtained from unheated and heated SIF (**top**), GIF (**middle**), and CIF (**bottom**) during gastric digestion in IHGS at different times. CM: cow milk.

**Table 1 foods-13-01286-t001:** Information of the infant formulae.

Average per 100 g	CIF	GIF	SIF
Energy (kJ)	2043	2073	2045
Protein (g)	15.30 ± 0.17	13.13 ± 0.06	12.76 ± 0.10
Fat (g)	24.43 ± 2.16	27.43 ± 2.64	22.64 ± 1.52
Carbohydrate (g)	52	54	52

## Data Availability

The original contributions presented in the study are included in the article, further inquiries can be directed to the corresponding authors.

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
