# Peer review of "Dynamic In Vitro Gastric Digestion Behaviour of Commercial Infant Formulae Made with Cow, Goat and Sheep Milk"

_foods, 2024, doi:10.3390/foods13091286_

Round 1
Reviewer 1 Report
Comments and Suggestions for Authors
Review of the Article: Dynamic in-vitro Gastric Digestion Behaviour of Commercial Infant Formulae Made with Cow, Goat and Sheep Milk
The reviewed manuscript compares the effects of protein composition and heat treatment on the in vitro gastric digestion and coagulation behavior of commercial infant starter formulas made from cow, goat, and sheep milk, using a dynamic infant human stomach simulator. The work is quite interesting, although the addressed topic is not quite novel and several similar studies can be found, some of them are cited by authors (not citated by authors but very interesting in this topic is: He, T., Rombouts, W., Einerhand, A. W. C., Hotrum, N., & van de Velde, F. (2022). Gastric protein digestion of goat and cow milk infant formula and human milk under simulated infant conditions. International Journal of Food Sciences and Nutrition, 73(1), 28–38. https://doi.org/10.1080/09637486.2021.1921705.).
The strength of the manuscript is the comprehensive comparison (same analytical model, same dynamic human infant stomach simulator) of three commercial formulations from three different animal species, each with a distinct protein profile affecting functional properties, including their effectiveness for infant consumption. Worth to emphasize is the very well-prepared Discussion section.
The manuscript is adequately prepared, nevertheless contains some shortcomings that need improvement before publication. Criticisms mainly revolve around the lack of results from densitometric analysis. In the section Materials and methods 2.7 SDS- PAGE it clearly described that gels were underwent such analysis (line 199-200), but the results have not be presented. In the Result section the SDS-PAGE results are presented on Figures (2 and 6) and discussed only on accompanying images.
However, quantitative interpretation of the results was mentioned, what is suggesting a need for use of densitometric analysis, to enable objective numerical comparisons. For instance, discussing protein digestibility (Figure 6), only band intensities were compared (e.g., "casein bands reduced gradually" at line 423), while these differences could be described precisely by providing values (e.g., in %). Similarly, discussion concerning results presented on the Figure 1 (lines 223-226, 233-235). Before publication the improvements are required.
On Figures 2, 4, 5 are lost the explanation of meaning letters a,b,c..
Author Response
The reviewed manuscript compares the effects of protein composition and heat treatment on the in vitro gastric digestion and coagulation behavior of commercial infant starter formulas made from cow, goat, and sheep milk, using a dynamic infant human stomach simulator. The work is quite interesting, although the addressed topic is not quite novel and several similar studies can be found, some of them are cited by authors (not citated by authors but very interesting in this topic is: He, T., Rombouts, W., Einerhand, A. W. C., Hotrum, N., & van de Velde, F. (2022). Gastric protein digestion of goat and cow milk infant formula and human milk under simulated infant conditions. International Journal of Food Sciences and Nutrition, 73(1), 28–38. https://doi.org/10.1080/09637486.2021.1921705.).
AU: The mentioned references has been added to our reference list in the revised version.
The strength of the manuscript is the comprehensive comparison (same analytical model, same dynamic human infant stomach simulator) of three commercial formulations from three different animal species, each with a distinct protein profile affecting functional properties, including their effectiveness for infant consumption. Worth to emphasize is the very well-prepared Discussion section.
AU: thanks for the positive comments.
The manuscript is adequately prepared, nevertheless contains some shortcomings that need improvement before publication. Criticisms mainly revolve around the lack of results from densitometric analysis. In the section Materials and methods 2.7 SDS- PAGE it clearly described that gels were underwent such analysis (line 199-200), but the results have not be presented. In the Result section the SDS-PAGE results are presented on Figures (2 and 6) and discussed only on accompanying images. However, quantitative interpretation of the results was mentioned, what is suggesting a need for use of densitometric analysis, to enable objective numerical comparisons. For instance, discussing protein digestibility (Figure 6), only band intensities were compared (e.g., "casein bands reduced gradually" at line 423), while these differences could be described precisely by providing values (e.g., in %). Similarly, discussion concerning results presented on the Figure 1 (lines 223-226, 233-235). Before publication the improvements are required.
AU: thanks for this valuable suggestion. We did the densitometric analysis of SDS-PAGE for individual protein hydrolysis in our previous report (Ye et al., 2019), but we found the accuracy of quantification based on the SDS-PAGE is a question, especially for the other species animal milks such as sheep milk, in which the caseins were difficult to be separated well. Its result is different compared with other techniques such as HPLC. Therefore, we will not present the densitometric analysis for quantification of individual protein in the present manuscript. We hope that they could be presented and discussed in other opportunity in future.
On Figures 2, 4, 5 are lost the explanation of meaning letters a,b,c.
AU: the explanations of letters a, b, c have been added in the captions of these Figures in the revised manuscript.
Reviewer 2 Report
Comments and Suggestions for Authors
Dear Authors
the paper is well presented and organized. I suggested the authors some minor revisions to it, however I wish to inform you that there is a 1:3 ratio of self-references vs total references, which is quite high
your manuscript is well written and organized, bringing some interesting information on the topic of milk protein digestion. I suggest you some minor revisions in the attached file.
Best regards

Author Response
The manuscript is well presented and organized, and I suggest the following minor revisions:
Lines 156, 172, 227 are a bit unclear, could you please revise them?
AU: they are modified in the revision.
Line 175: Could you explain what is intended here with “gels”?
AU: this sentence has been modified (L175).
Line 196 and followings: please revise the sentence. What is the acrylamide percentage of the gels?
AU: it is 15% acrylamide gel. It is clarified in revision (L188).
Figure 6: please add the markers for molecular weight in the figure. I suggest also to highlight on the
gels the 15 kDa band you talk about in line 395.
AU: there is the protein patterns of milk as control in each PAGE. We consider the milk protein is the best control marker for this experiment as the molecular weight of proteins from milk are well known. Therefore, we don’t not add the markers of molecular weight in the figure. In addition, we can refer the Fig. 1 where there are molecular weight markers if it is necessary. We put the indication arrow for 15 kDa band in figure in the revision.
Line 390 and followings: without the reference of a protein marker it is hard to compare the protein profiles, since unheated GIF and CIF seem to have run for a shorter time compared to the heated ones. Taking into account these differences, the k-casein band seems still visible in unheated GIF and CIF samples, so I suggest to higlight on the figure (e.g. with arrows) the bands that change in intensity.
AU: regarding the reference of a protein marker, please see the answer for last comment. The position of k-casein band has been pointed out in Figure 6. We don’t agree with the reviewer’s comment on “the k-casein band seems still visible in unheated GIF and CIF samples”, the k-casein band was not observed from both unheated and heated samples from 20 to 160 min. Therefore, we can not highlight the changes in its band in digested samples in Fig.
Please revise the sentences in lines 243, 364, 367-369, 438-439, 508, 513-514, 522-525
AU: some of these sentences have been modified.
There are some typos in lines 235, 249, 292, 306, 472
AU: modified.
Line 500: “hides”
AU: changed
Missing spaces between words in lines 423, 472, 512
AU: Corrected.
I suggest to include other references on the topic, to decrease the ratio of self-references.
AU: some other references have been added to the revision.
Reviewer 3 Report
Comments and Suggestions for Authors
Reviewer’s comments
Title : It seems to be suitable for the scope of journal area. However, in my opinion, “Dynamic in vitro……, Goat and Sheep Milk” should be “Dynamic in vitro……, Goat or Sheep Milk”.
Abstract : It seems to be O.K. for me. The abstract shows the objective of the study on that title quite clearly. The results of this study in abstract part are a main finding but the novelty should be mentioned.
Keywords : It is O.K. and eases to help the reader understand and can find in the scientific system searching.
Introduction : The content is fine and quite updated for previous work relevant to the present work. However, the authors should aware that the English grammar should be edited such as the sentences on Line 42-43 and 51-53, as well as the correction of typing e.g., spaces between words on Line 37. The abbreviation should be used after that word was mentioned as a full word at the first time e.g., the abbreviations on Line 73-75, as well as in content of text which can be representative of those abbreviations in Table 1. Moreover, the authors should give the definition or information of “dynamic gastric digestion” to let the boarder readers understand.
Materials and methods :
Line 120, please check for the English grammatical errors.
Table 1, Is there a significant difference between energies and carbohydrates of those IFs?
Line 126, Fat content in each IF seemed to be different. This might affect the results, especially the result of fat particle distribution and size. Please include this notice in your discussion in that part to make your manuscript more reasonable.
Line 133-143, The clarity of those sentences needs to be revised. It is better to specify how many units of the activity of pepsin was used in this experiment than mentioning a general recommendation for the activity of that enzyme based on weight of infant. The question for this point is what the weight of infant the authors used or fixed in this study.
Line 138, What does “by two internal heaters and a thermostat” mean?
Line 152-158, What is the temperature used for pH measurement?
Line 159-170, What is the substrate that the authors measure in this part? Fat or any suspended particles (such as protein). This part needs to be clarified. The refractive indices are specified for what substrate, please explain. Do not forget that each IF contained quite different fat content as mentioned in Line 126, thus the authors should realize this point in this part too.
Line 175, What is the combination gel? Did the authors mention gel before this part?
Line 187, To write the detail of equipment used in the experiment in scientific manuscript, Model, Company name, City, and Country should be written too.
Line 198-199, 206, There is an English grammatical error.
Line 199-200, The software needs to be quoted from the software company and follow the scientific quotation as mentioned in the comment on Line 187.
Line 208-209, What does the “repleted-measures two-actor ANOVA model” mean? Please clarify.
Line 215-216, The authors should add more detail as mentioned in the comment on Line 187.
The least is about the consistency of writing words e.g., unit (min or minute, h or hour, in vitro or in vitro, ml or mL), please carefully check those words.
Results :
Since this work seems to be very complicated, the authors should explain step by step the results that must be relevant to the step of methodology precisely.
Line 228, alpha s1 casein does not be found in SIF too. Why did the authors mention that “GIF showed the lowest content of alpha s1 casein”.
Line 231, When the authors reported or quoted the results or the other works in percentage, the results of the authors should be in percentage too.
Line 261-263, How could the open structure of heated whey protein reduce the pH of IF? Please explain more at this point.
Figure 2, the rationale for using protein concentration at 1.6% was not mentioned previously anywhere in the manuscript until its appearance was on Line 479. Please revise this point.
Line 269, Please specify which type of those milk proteins coagulate at pH 6.2-6.3?
Line 306, extra ).
Line 334-340, Needs more explanation or discussion at this point why the results came up like that. If the reasons for these results were mentioned in Line 485-514, please quote the results which were found in Line 334-340 to make the readers easier to understand clearly.
Line 346, What are the “expected values”? Please clarify.
Line 351-352, English grammatical error should be checked.
Figure 4 and 5, What do the small letters present on the lines in each graph? Please explain.
Figure 4, When the authors said the particle size of chyme, what that particle size should be a representative of fat or protein in those IFs. Please clarify and might give more information about this in the methodology part.
Figure 5, What is dilution line? Please explain and try to add the meaning of that dilution to the content in the manuscript too, otherwise the readers may be confused.
Line 358, Protein content in SIF seems to exhibit no difference in each time of digestion, right? Please check.
Line 361, “the protein content of”
Line 382-387, please explain more at this point. If the authors explained in a discussion part, the authors should try to link consequently results and discussion clearly to make it easier for the readers to follow those discussions by quoting to the results e.g., the number of what was gained in the results.
Line 406-408, That is the assumption of the raw material preparation, however, the authors should reveal the correlation between that assumption and the results more evidently.
Line 423, space needs
Line 469-471, English grammatical error
Line 492-494, Please rewrite the sentence to show the sequence of unheated to heated IF, especially the effect of heat on protein denatures which might impact the releasing of fat in that chyme. In my opinion, I suggest that the authors should specify what the particle size was (fat or protein) and try to explain gradually and carefully from one stage to the next of those happenings related to each component of fat and protein.
Line 512, What is “ratein”?
Conclusion :
Line 521, The authors did not do the experiment about the effect of homogenization in this study, in my opinion, I think the elimination of homogenization should be applied here.
The conclusion does not seem to meet the objectives of this study yet since there is no mention of dynamic digestion at all.
Comments on the Quality of English LanguageThere are many grammatical errors in this manuscript. The authors should edit the manuscript by using a native speaker or proofreading service center.
The authors should also be aware of making spaces between words and using parenthesis.
Author Response
Title : It seems to be suitable for the scope of journal area. However, in my opinion, “Dynamic in vitro……, Goat and Sheep Milk” should be “Dynamic in vitro……, Goat or Sheep Milk”.
AU: thanks, changed.
Abstract : It seems to be O.K. for me. The abstract shows the objective of the study on that title quite clearly. The results of this study in abstract part are a main finding but the novelty should be mentioned.
AU: thanks, the novelty is working on commercial infant formulae which have not been well studied which has been mentioned in abstract.
Keywords : It is O.K. and eases to help the reader understand and can find in the scientific system searching.
AU: thanks.
Introduction : The content is fine and quite updated for previous work relevant to the present work. However, the authors should aware that the English grammar should be edited such as the sentences on Line 42-43 and 51-53, as well as the correction of typing e.g., spaces between words on Line 37. The abbreviation should be used after that word was mentioned as a full word at the first time e.g., the abbreviations on Line 73-75, as well as in content of text which can be representative of those abbreviations in Table 1. Moreover, the authors should give the definition or information of “dynamic gastric digestion” to let the boarder readers understand.
AU: they have been corrected in revision.
Materials and methods :
Line 120, please check for the English grammatical errors.
AU: corrected.
Table 1, Is there a significant difference between energies and carbohydrates of those IFs?
AU: No, they are not significant as they are produced by same company.
Line 126, Fat content in each IF seemed to be different. This might affect the results, especially the result of fat particle distribution and size. Please include this notice in your discussion in that part to make your manuscript more reasonable.
AU: thanks for this comment. The effect of this differences in fat content and fat globule size on the coagulation and digestion of protein were limited.
Line 133-143, The clarity of those sentences needs to be revised. It is better to specify how many units of the activity of pepsin was used in this experiment than mentioning a general recommendation for the activity of that enzyme based on weight of infant. The question for this point is what the weight of infant the authors used or fixed in this study.
AU: 7.5 kg for 6-month-old infants in the manuscript (L134).
Line 138, What does “by two internal heaters and a thermostat” mean?
AU: heaters and thermostat located at inside of the IHGS model.
Line 152-158, What is the temperature used for pH measurement?
AU: samples’ temperature is 37⁰C.
Line 159-170, What is the substrate that the authors measure in this part? Fat or any suspended particles (such as protein). This part needs to be clarified. The refractive indices are specified for what substrate, please explain. Do not forget that each IF contained quite different fat content as mentioned in Line 126, thus the authors should realize this point in this part too.
AU: the size of fat globule/oil droplet in IFs was measured in here. IF can be considered as a food emulsion system. However, the flocculation of fat/oil droplets was induced by protein on surface. The flocculated fat globules/oil droplets include some protein at interface. Refractive indices of 1.46 (with an absorbance value of 0.001) and 1.33 were used for dispersed phase and aqueous phase, respectively was typical for emulsion measurement. The difference in fat content has limited on the effect on size measurement.
Line 175, What is the combination gel? Did the authors mention gel before this part?
AU: thanks for the question. This sentence has been modified in revision.
Line 187, To write the detail of equipment used in the experiment in scientific manuscript, Model, Company name, City, and Country should be written too.
AU: added.
Line 198-199, 206, There is an English grammatical error.
AU: corrected.
Line 199-200, The software needs to be quoted from the software company and follow the scientific quotation as mentioned in the comment on Line 187.
AU: Bio-Rad Image Lab software version 6.1.
Line 208-209, What does the “repleted-measures two-actor ANOVA model” mean? Please clarify.
AU: A two-way repeated measures ANOVA compares the mean differences between groups that have been split on two within-subjects factors (also known as independent variables). A two-way repeated measures ANOVA is often used in studies where you have measured a dependent variable over two or more time points, or when subjects have undergone two or more conditions (i.e., the two factors are "time" and "conditions"). "Repeated measures" means that one of the factors was repeated. For example, here we compare three IF samples, and measure each subject at several time points (repeated).
Line 215-216, The authors should add more detail as mentioned in the comment on Line 187.
AU: we are not sure what reviewer required to be added.
The least is about the consistency of writing words e.g., unit (min or minute, h or hour, in vitro or in vitro, ml or mL), please carefully check those words.
AU: thanks, edition is done.
Results :
Since this work seems to be very complicated, the authors should explain step by step the results that must be relevant to the step of methodology precisely.
Line 228, alpha s1 casein does not be found in SIF too. Why did the authors mention that “GIF showed the lowest content of alpha s1 casein”.
AU: not sure where it was. we said “there is a lower level alpha S1-casein in goat milk than in other ruminants’ milk” from cited references.
Line 231, When the authors reported or quoted the results or the other works in percentage, the results of the authors should be in percentage too.
AU: we did not quantify the casein concentration of IFs in the present report.
Line 261-263, How could the open structure of heated whey protein reduce the pH of IF? Please explain more at this point.
AU: the sentence has been modified in revision (L259-262). “The slow decrease in pH of heated milk could be related to the denaturation of whey protein associated with casein micelles, forming more open structured aggregates under gastric conditions” which has been reported in previous study (Ye et al., 2016).
Figure 2, the rationale for using protein concentration at 1.6% was not mentioned previously anywhere in the manuscript until its appearance was on Line 479. Please revise this point.
AU: it was mentioned in section 2.2 Preparation of liquid infant formulae samples (L120).
Line 269, Please specify which type of those milk proteins coagulate at pH 6.2-6.3?
AU: casein micelles coagulated induced by pepsin at about pH 6. This has been reported and discussed in previous reports (Ye et al., 2017, 2019, Pan et al., 2021, Li et al., 2022). These have been added to discussion section (L455-457).
Line 306, extra ).
AU: ) has been removed.
Line 334-340, Needs more explanation or discussion at this point why the results came up like that. If the reasons for these results were mentioned in Line 485-514, please quote the results which were found in Line 334-340 to make the readers easier to understand clearly.
AU: thanks for this. The results in Fig. 4 has been cited in the discussion section in revision (L500 and L502).
Line 346, What are the “expected values”? Please clarify.
AU: following the dilution line was expected values as stated in manuscript “the expected values based on dilution”.
Line 351-352, English grammatical error should be checked.
AU: done.
Figure 4 and 5, What do the small letters present on the lines in each graph? Please explain.
AU: the explanations have added to figures’ captions. Please see L314-317 and L358-362.
Figure 4, When the authors said the particle size of chyme, what that particle size should be a representative of fat or protein in those IFs. Please clarify and might give more information about this in the methodology part.
AU: the answer has been responded and addressed in former similar question.
Figure 5, What is dilution line? Please explain and try to add the meaning of that dilution to the content in the manuscript too, otherwise the readers may be confused.
AU: the explanation has been added to the revised version. Please see L345-349.
Line 358, Protein content in SIF seems to exhibit no difference in each time of digestion, right? Please check.
AU: SIF protein contents exhibited differences at certain time points (Figure 5 (A-P).
Line 361, “the protein content of”
AU: “protein” was added in revision.
Line 382-387, please explain more at this point. If the authors explained in a discussion part, the authors should try to link consequently results and discussion clearly to make it easier for the readers to follow those discussions by quoting to the results e.g., the number of what was gained in the results.
AU: these sentences have been modified in revision.
Line 406-408, That is the assumption of the raw material preparation, however, the authors should reveal the correlation between that assumption and the results more evidently.
AU: thanks for this comment. authors had the tests on WPNI for denaturation levels of IFs for testing this assumption. However, we do not show this in the present paper.
Line 423, space needs
AU: done.
Line 469-471, English grammatical error
AU: done.
Line 492-494, Please rewrite the sentence to show the sequence of unheated to heated IF, especially the effect of heat on protein denatures which might impact the releasing of fat in that chyme. In my opinion, I suggest that the authors should specify what the particle size was (fat or protein) and try to explain gradually and carefully from one stage to the next of those happenings related to each component of fat and protein.
AU: thanks for this suggestion. Some modifications on these sentences have been done in the revised version. The specification in the results of fat or protein behaviours of IFs have been described in the result section, here is only the discussion on the possible reason for the results.
Line 512, What is “ratein”?
AU: change to “rate in”.
Conclusion :
Line 521, The authors did not do the experiment about the effect of homogenization in this study, in my opinion, I think the elimination of homogenization should be applied here.
AU: homogenization” is removed from revision.
The conclusion does not seem to meet the objectives of this study yet since there is no mention of dynamic digestion at all.
AU: “dynamic in vitro” has been added to conclusion section (see L530).
Round 2
Reviewer 1 Report
Comments and Suggestions for Authors
Most of the comments were considered by Authors, although one important (concerning densytometric analysis) wasn't. I don't agree with the authors' explanation for not including the results of the densytometric analysis when comparing the amount of particular protein fractions. I would like to point out that the use of correct softwareto measure optical density of protein bands, which as described in "Materials and methods" was the case, makes it possible to determine accurately the differences in the intensity of the protein bands, especially when the samples are on the same gel . I absolutely agree that the results will not be as accurate as after HPLC analysis, but still more accurate, and scientifically sounded than the laconic descriptions provided by authors. Of course even the description posted by authors are not very accurate still indicates observed trends. Taking the above into account, I leave the final decision to accept or reject a manuscript to the Editor.
Author Response
Most of the comments were considered by Authors, although one important (concerning densytometric analysis) wasn't. I don't agree with the authors' explanation for not including the results of the densytometric analysis when comparing the amount of particular protein fractions. I would like to point out that the use of correct softwareto measure optical density of protein bands, which as described in "Materials and methods" was the case, makes it possible to determine accurately the differences in the intensity of the protein bands, especially when the samples are on the same gel . I absolutely agree that the results will not be as accurate as after HPLC analysis, but still more accurate, and scientifically sounded than the laconic descriptions provided by authors. Of course even the description posted by authors are not very accurate still indicates observed trends. Taking the above into account, I leave the final decision to accept or reject a manuscript to the Editor.
AU: thanks again for the reviewer’s suggestion. Despite the reviewer considers that the results of the densytometic analysis are better to indicate the observed trends. We think that the description provided in the manuscript has already indicated the observed trends of protein changes during digestion. We will remove the method of densytometric analysis from materials and methods section.
Reviewer 3 Report
Comments and Suggestions for Authors
Line 37, needs space between benefits(Balthazar,…
In the introduction part, I have not seen any explanation of the definition of dynamic gastric digestion yet as I requested in the first-round review.
From my previous question in the first-round review as “Is there a significant difference between energies and carbohydrates of those IFs?” and the authors answered that “No, they are not significant as they are produced by same company.” I disagree with their answer since the number in Table 1 for carbohydrates seemed to be statistically significant difference.
Regarding my question as “Fat content in each IF seemed to be different. This might affect the results, especially the result of fat particle distribution and size. Please include this notice in your discussion in that part to make your manuscript more reasonable.” The authors answered that “The effect of this differences in fat content and fat globule size on the coagulation and digestion of protein were limited.” This is not clear how those matters were limited. What does that mean? The fat content in SIF, GIF, and CIF was 3.2, 3.3, and 2.4 g/ 100 g of IFs.
There is still a lack of equipment information such as pH meter. There is no detail of that pH meter for its company, city, and country provided. Also, the temperature for pH measurement should be mentioned there. Please check for the other equipment or software such as
Bio-Rad Image Lab software version 6.1 too. These need to add company, city, and country too.
As the authors answered that” The difference in fat content has limited on the effect on size measurement”. I totally disagree with that reason. The more amount of fat content found in CIF should provide more occasion to collide or coagulate with each other easier than the small amount of fat content in SIF and GIF.
In vitro should be written in italic form.
Consistency of writing or typing is still concerned such as mL. The authors used both ml and mL.
Please check unit used in the manuscript which should be followed the journal format such as hour should be h.
There is still no an explanation of the letters on lines in Figure 2. What do a, b, c, d, e, f, x, y, etc. mean?
Regarding the 1st-round comment “alpha s1 casein does not be found in SIF too. Why did the authors mention that “GIF showed the lowest content of alpha s1 casein”. The authors answered that “there is a lower level alpha S1-casein in goat milk than in other ruminants’ milk” from cited references. In my opinion, GIF and SIF did not find alpha s1, right? So, the authors should not write as “there is a lower level alpha S1-casein in goat milk than in other ruminants’ milk”, shouldn’t they?
Line 251-253, please rewrite a new sentence since I do not know what a thing that the authors wrote (pH or what)… “of heated SIF dropped from 4.85 ± 0.11 to 2.87 ± 0.17, while 252
unheated SIF decreased from 4.20 ± 0.19 to 2.59 ± 0.23.” Is it pH?
How can the denaturation of whey protein associated with casein micelles, forming more open structured aggregates under gastric conditions, decrease pH of this sample? Please explain based on food chemistry point of view.
Line 350, please make a new paragraph with the same tab as previous paragraph.
Line 435, need space “casein bandsreduced in intensity gradually,”
Comments on the Quality of English Languagejust some sentences that might be modified to meet the correct English grammar such as “In addition, the fat contents of emptied digesta of heated IFs were lower than that of unheated IFs, but were higher than that of unheated IFs at the late stage of digestion.”
Author Response
Line 37, needs space between benefits(Balthazar,…
AU: done.
In the introduction part, I have not seen any explanation of the definition of dynamic gastric digestion yet as I requested in the first-round review.
AU: “dynamic in vitro gastric digestion” in the present study was using human gastric simulator (HGS) to investigate the in vitro gastric digestion of IF. The dynamic gastric digestion model HGS incorporates gastric secretion, emptying systems, and temperature control that enable accurate simulation of dynamic digestion process for detailed investigation of the changes in the physical chemical properties of ingested foods. The simulated gastric contraction force demonstrates a similar pattern as in vivo stomach forces. The precise control of gastric secretion and emptying and the adjustable mechanical forces in the HGS provide a useful tool to study transformation of food constituents under simulated physiological conditions. The dynamic HDS model is relatively compared to static digestion model and simi-dynamic digestion model. This information has been described in huge amounts of previous papers. We don’t repeat the basic information again in this present manuscript.
From my previous question in the first-round review as “Is there a significant difference between energies and carbohydrates of those IFs?” and the authors answered that “No, they are not significant as they are produced by same company.” I disagree with their answer since the number in Table 1 for carbohydrates seemed to be statistically significant difference.
AU: Table 1 did not show they are statistically significant difference.
Regarding my question as “Fat content in each IF seemed to be different. This might affect the results, especially the result of fat particle distribution and size. Please include this notice in your discussion in that part to make your manuscript more reasonable.” The authors answered that “The effect of this differences in fat content and fat globule size on the coagulation and digestion of protein were limited.” This is not clear how those matters were limited. What does that mean? The fat content in SIF, GIF, and CIF was 3.2, 3.3, and 2.4 g/ 100 g of IFs.
AU: that means that The effect of this differences in fat content and fat globule size on the coagulation and digestion of protein were limited””. Please note the effect was limited. We consider this statement is right.
There is still a lack of equipment information such as pH meter. There is no detail of that pH meter for its company, city, and country provided. Also, the temperature for pH measurement should be mentioned there. Please check for the other equipment or software such as
Bio-Rad Image Lab software version 6.1 too. These need to add company, city, and country too.
AU: added.
As the authors answered that” The difference in fat content has limited on the effect on size measurement”. I totally disagree with that reason. The more amount of fat content found in CIF should provide more occasion to collide or coagulate with each other easier than the small amount of fat content in SIF and GIF.
AU: reviewer should note the difference in fat content between IFs is not that big different, in addition, the sample should be diluted to a same concentration during measurement of size using MasterSizer.
In vitro should be written in italic form.
AU: Done.
Consistency of writing or typing is still concerned such as mL. The authors used both ml and mL.
AU: has checked through out the manuscript and will do again during proof.
Please check unit used in the manuscript which should be followed the journal format such as hour should be h.
AU: done.
There is still no an explanation of the letters on lines in Figure 2. What do a, b, c, d, e, f, x, y, etc. mean?
AU: the statement is added to Figure 2 caption.
Regarding the 1st-round comment “alpha s1 casein does not be found in SIF too. Why did the authors mention that “GIF showed the lowest content of alpha s1 casein”. The authors answered that “there is a lower level alpha S1-casein in goat milk than in other ruminants’ milk” from cited references. In my opinion, GIF and SIF did not find alpha s1, right? So, the authors should not write as “there is a lower level alpha S1-casein in goat milk than in other ruminants’ milk”, shouldn’t they?
AU: We understand the reviewer’s concern. However, it is not such simple, we did not say “ GIF and SIF did not find alpha s1”, although there is not clear separated alpha S1 and beta casein in SIF in the present manuscript, it is because SDS-PAGE is difficult to separate them of sheep milk. In fact, there are alpha S1 casein in SIF if using other technique to determine the composition of protein in sheep milk. Therefore, we said there is a lower level alpha S1-casein in “GIF than in other ruminants’ milk IFs” is safer.
Line 251-253, please rewrite a new sentence since I do not know what a thing that the authors wrote (pH or what)… “of heated SIF dropped from 4.85 ± 0.11 to 2.87 ± 0.17, while 252 unheated SIF decreased from 4.20 ± 0.19 to 2.59 ± 0.23.” Is it pH?
AU: yes, pH. The sentence has been modified.
How can the denaturation of whey protein associated with casein micelles, forming more open structured aggregates under gastric conditions, decrease pH of this sample? Please explain based on food chemistry point of view.
AU: decrease pH of the samples was because the aggregate with open structure allows more gastric fluid (with very low pH) easily diffuse and migrate into the inside of aggregate, which might lead to quick decrease in pH of sample, which has been reported and explained in our previous reports (Ye et al., 2016).
Line 350, please make a new paragraph with the same tab as previous paragraph.
AU: done.
Line 435, need space “casein bandsreduced in intensity gradually,”
AU: done.
Comments on the Quality of English Language just some sentences that might be modified to meet the correct English grammar such as “In addition, the fat contents of emptied digesta of heated IFs were lower than that of unheated IFs, but were higher than that of unheated IFs at the late stage of digestion.”
AU: this sentence has been modified.